# In Vitro Anti-Influenza Virus Activity of Non-Polar *Primula veris* subsp. *veris* Extract

**DOI:** 10.3390/ph15121513

**Published:** 2022-12-05

**Authors:** Aristides G. Eliopoulos, Apostolis Angelis, Anastasia Liakakou, Leandros A. Skaltsounis

**Affiliations:** 1Department of Biology, School of Medicine, National and Kapodistrian University of Athens, 11527 Athens, Greece; 2Center of Basic Research, Biomedical Research Foundation of the Academy of Athens, 11527 Athens, Greece; 3Department of Pharmacy, Division of Pharmacognosy and Natural Products Chemistry, National and Kapodistrian University of Athens, 15771 Athens, Greece

**Keywords:** *Primula veris*, influenza, *Cistus creticus*, *Echinaceae purpurea*, supercritical extraction

## Abstract

Medicinal plants have long been recognized as a tremendous source of candidate compounds for the development of pharmaceuticals, including anti-viral agents. Herein, we report the identification of anti-influenza virus activity in non-polar *Primula veris* L. subsp. *veris* extracts. We show that *P. veris* subsp. *veris* flower extracts, obtained using supercritical fluid or ultrasound-based extraction, possess virucidal/virus inactivation properties and confer prophylactic and therapeutic effects against influenza virus-induced cytolysis in vitro. By GC-MS and UPLC-HRMS analysis of non-polar *P. veris* subsp. *veris* extracts we identified terpenes, flavones, tocopherols, and other classes of phytochemicals with known or putative anti-influenza properties. In silico prediction of cellular functions and molecular pathways affected by these phytochemicals suggests putative effects on signal transduction, inflammasome, and cell death pathways that are relevant to influenza virus pathogenesis. Combining *P. veris* subsp. *veris* with extracts of medicinal plants with proven anti-influenza activity such as *Echinacea purpurea* (L.) Moench and *Cistus creticus* L. subsp. *creticus* achieves an impressive protective effect against infection by influenza virus H1N1 in vitro and reduced progeny virus production by infected cells. Collectively, these findings uncover a previously uncharted biological property of non-polar *P. veris* flower extracts that warrants further studies to assess clinical efficacy.

## 1. Introduction

Influenza viruses belong to the family of *Orthomyxoviridae*. Their genome comprises an eight-segment, negative-sense RNA that is prone to mutations as it lacks proofreading mechanisms during replication. The virus genome encodes surface glycoproteins hemagglutinin (HA) and neuraminidase (NA), which play essential roles in virus life cycle and virulence. HA enables virus attachment to cell surface receptors and internalization and NA is essential for virion release from infected cells. Influenza viruses replicate predominantly in the airway epithelia, leading to inflammation, airway congestion, and tissue necrosis. Illness is manifested by the involvement of the lower respiratory tract (cough), accompanied with headache, myalgia, and fever, but in a subset of patients it may progress to severe complications that include pneumonia, encephalitis, myocarditis, and myelitis, and may lead to death [1].

Currently, influenza virus therapeutics are largely limited to neuraminidase inhibitors, such as oseltamivir, zanamivir, and peramivir [2]. A new drug, baloxavir marboxil, which inhibits viral cap-dependent endonuclease activity, was developed and approved in 2018 by the U.S. Food and Drug Administration to enrich the clinical armamentarium against influenza viruses [3]. However, seasonal influenza viruses have been found to be resistant to oseltamivir, raising concerns about the long-term availability of therapeutic agents to tackle the pathological manifestations of virus infection [4]. Therefore, the development of novel anti-influenza drugs to control future influenza epidemics is warranted.

Traditional medicinal plants have long been recognized as a major source of candidate compounds for the development of pharmaceuticals, with as much as 40% of modern drugs being derived, directly or indirectly, from natural sources [5,6,7]. This also applies to anti-viral agents. Oseltamivir, for example, which is sold under the brand name Tamiflu to treat symptoms caused by influenza, was synthesized from shikimic acid extracted from *Illicium verum*, the Chinese star anise. Empirical knowledge of health benefits motivates the search for plant extracts and functional phytochemicals for anti-influenza activity. Thus far, several natural products and extracts from medicinal plants have been reported to possess anti-viral activity, which has led to formulations such as Sinupret^®^, a herbal medicinal product made from Gentian root, Primula flower, Elder flower, Sorrel herb, and Verbena herb [8]; and Echinaforce^®^, a standardized preparation derived by ethanol extraction of *Echinaceae purpurea* (L.) Moench herb and roots [9]; among others. Given the wealth of plant varieties, geographical influences on phytochemical composition and methods of extraction, the identification of anti-influenza activities in other plants and plant extracts is by no means a completed endeavor [7].

In Greece, an abundance of medicinal plants has been explored since ancient times to combat flu-like symptoms. *Cistus creticus* L. (Cistaceae), for example, is depicted in Minoan frescoes of c. 1550 B.C. and is still being used in Cretan villages as herbal decoction to alleviate flu symptoms and cough. By performing a small-scale screening of 41 extracts derived from nine plant species collected from Greece, we have identified novel anti-influenza virus activities in *Primula veris* L. subsp *veris* non-polar extracts. Previous studies have identified anti-inflammatory, anti-microbial, and anti-fungal effects in *Primulaceae* extracts [10,11], but anti-influenza virus properties have not been ascribed to Primula before. Additionally, we show that by combining *Primula veris* extracts with those of medicinal plants with proven anti-influenza activity, such as *Cistus creticus* subsp. *Creticus* and *Echinacea purpurea*, achieves an impressive protective effect against H1N1 in vitro.

## 2. Results

### 2.1. Extraction of P. veris subsp. veris Flowering Parts

The extraction of the flowering parts of *P. veris* subsp. *veris* was initially achieved by using an ultrasound extraction technique that offers a rapid and effective recovery of all substances from the plant material [12]. By initially extruding the raw material with dichloromethane (DCM) and then with methanol and water, three different extracts, containing the non-polar, medium polarity and polar compounds, respectively, were obtained. This extraction procedure should enable the establishment of direct associations between the anti-viral activity of an extract with the polarity of the secondary metabolites, thus facilitating future studies to identify active substances. The extraction of 200 g of plant material by the ultrasound extraction (UE) technique resulted in 3.64 g of DCM extract (1.82% *w/w*), 22.44 g of MeOH extract (11.22% *w/w*), and 32.33 g of H_2_O extract (16.16% *w/w*).

The extraction of *P. veris* subsp. *veris* flowering parts was repeated using supercritical fluid extraction, which is extensively used as a “green” technique that yields extracts free of toxic solvents [13]. Twenty grams (20 g) of dry powdered material were subjected to a continuous three-step extraction procedure. The extraction started by passing through the plant material only in supercritical CO_2_, followed by addition of 5% ethanol as co-solvent and, finally, passing a mixture of CO_2_ with 13% EtOH as extraction solvent. Each step lasted 90 min resulting in the recovery of 110mg of CO_2_ extract (0.55% *w/w*), 306.9 mg of CO_2_ + 5% EtOH extract (1.53% *w/w*), and 149.1 mg of CO_2_ + 13% EtOH extract (0.75% *w/w*). In all cases, the extracts were dried and re-dissolved in DMSO prior to the assessment of biological activity.

### 2.2. Identification of Primula veris Extracts with Virucidal Activity

We evaluated the putative virucidal/virus inactivation effects of *P. veris* subsp. *Veris* DCM, MeOH, or H_2_O extracts by incubating H1N1 virus with 50 μg/mL of each extract for 1 h prior to infection of MDCK cells at MOI = 0.1 (Figure 1A). Virus-induced cytotoxicity was assessed 24 to 30 h later by microscopic examination and quantification of cell survival by MTT conversion or LDH release assays.

Compared to uninfected cultures, exposure to the virus elicited cytotoxic effects that were microscopically evident by the accumulation of rounded cells (Figure 2A) and by an overall 40% reduction in the survival of MDCK cell population, assessed by MTT conversion (Figure 2B). However, incubation of H1N1 with DCM extracts of *P. veris* was found to reduce cytolysis by approximately 50% (Figure 2A,B). The MeOH extracts of *P. veris* were also found to mediate virus inactivation but to a lesser degree, whereas polar extracts in H_2_O had no effect (Figure 2B). Treatment with 50 μg/mL of each extract alone (i.e., in the absence of virus) did not impact MDCK cell survival.

The aforementioned data indicated that DCM-soluble components of *P. veris* have putative virucidal/virus inactivation effects. Thus, we tested if substitution of DCM with safer solvents could also enable extraction of virucidal *P. veris* substances. To this end, we applied extracts isolated by a supercritical CO_2_ fluid extraction method in the presence or absence of 5% or 13% EtOH as co-solvent and assessed virus inactivation vis-à-vis DCM *P. veris* extract.

Microscopically, the extracts retrieved by supercritical CO_2_ fluid extraction in the presence of either 5% or 13% EtOH were equally or more efficient than DCM extract in reducing the virus-induced accumulation of rounded cells (Figure 3A), indicative of significant virucidal/virus inactivation effects. These were quantified by measuring LDH released in the culture supernatant following MDCK infection with H1N1 or with virus that had been pre-incubated for 1 h with 10, 50 or 100 μg/mL of *P. veris* extracts. The results (Figure 3B) confirmed the microscopy observations and registered a concentration-dependent decrease in virus-induced cytotoxicity by supercritical CO_2_ in the presence of 5% or 13% EtOH that exceeded the effects of DCM extracts.

On the basis of the aforementioned results, we focused our subsequent analyses on *P. veris* flower extracts isolated by supercritical CO_2_ in the presence of 13% ethanol (hereafter, “Primula”) to test for putative anti-viral effects under different modes of treatment, namely prophylactic administration and therapeutic administration (Figure 1B,C). We also compared the anti-viral effects of Primula to those of extracts from medicinal plants with proven anti-influenza activity, namely *E. purpurea* [9,14] and *C. creticus* [15], and investigated putative additive activities when combined.

### 2.3. Prophylactic Effects of Primula Plant Extracts against Influenza Virus-Induced Cell Lysis

To identify potential prophylactic effects of Primula on influenza virus-induced cytotoxicity, MDCK cells were treated with 50 μg/mL of Primula extract for 4 h, washed, infected with H1N1 and then cultured in standard growth media. Cell lysis was quantified 24 to 30 h later by measuring LDH released in the culture supernatant. As shown in Figure 4, Primula extracts reduced virus-induced cytotoxicity by approximately 65%.

We also tested several Cistus and Echinacea extracts recovered by using different solvents and identified hydroethanolic *C. creticus* and EtOH/H_2_O-Xad7-derived *E. purpurea* extracts as the most potent in reducing virus-induced cytotoxicity (Figure 4). Combinations of either of these extracts with Primula did not augment protection in a statistically significant manner but triple combination significantly enhanced protection from virus-induced cytolysis (*p* < 0.05), indicative of complementary mechanisms in prophylactic anti-viral response.

### 2.4. Therapeutic Effects of Primula Extract against Influenza Virus-Induced Cell Lysis

To identify potential therapeutic effects of Primula extracts on influenza virus-induced cytotoxicity, MDCK cells were infected with H1N1 and then cultured in the presence or absence of 25 or 50 μg/mL of Primula. Microscopically, Primula had a profound effect on reducing virus-induced accumulation of rounded cells (Figure 5A), indicative of protective effects. These were quantified by measuring LDH released in the culture supernatant as a read-out for virus-induced cell lysis. The results (Figure 5A,B) confirmed the microscopy observations and registered a concentration-dependent decrease in virus-induced cell lysis by each extract that ranged between 40–60%. Oseltamivir (100 μg/mL) was used as positive control for inhibition of cytopathic effects demonstrating > 96% reduction in H1N1-mediated cell lysis.

Primula was found to be more effective in combination with Cistus rather than Echinacea and the triple combination did not offer additional therapeutic outcome. Cistus in combination with Echinacea resulted in increased therapeutic effect compared to each extract alone (Figure 5A–C). Therefore, Cistus in combination with either Primula or Echinacea achieves major therapeutic effects on influenza virus-induced cell lysis.

### 2.5. Primula, Cistus, and Echinacea Combination Treatment Results in Reduced Progeny Virus Production

We next combined the aforementioned administration protocols to assess the impact of the triple combination of Primula, Cistus, and Echinacea on influenza virus propagation. MDCK cell cultures were infected with H1N1 at MOI = 0.01 in the absence or presence of triple combination of plant extracts before, during, and after infection. In both cases, supernatants were assayed for progeny virus yields 24 h post-infection by standard plaque titrations. The results showed significant (>85%) reduction in progeny virus titers by the triple combination of Primula, Cistus, and Echinacea (Figure 6).

### 2.6. Chemical Composition of Primula veris Non-Polar Extracts

The biologically active non-polar extracts of *P. veris* subsp. *veris* were analyzed for their chemical composition. Using GC-MS, both the crude DCM extract and the extract obtained after esterification were first analyzed (Appendix A). The results identified a plethora of fatty compounds including n-alkanes, saturated and unsaturated fatty acids, fatty alcohols and fatty ketones (Appendix A). The presence of the above fatty compounds in high amounts in the extract makes difficult the identification of non-fatty secondary metabolites, especially when using LC-MS. In order to overcome this problem and to achieve a deeper analysis of the non-polar extract, a fractionation step was introduced, followed by LC-MS analysis of each separated fraction.

The fractionation of the DCM extract was achieved by using countercurrent partition chromatography (CPC) technology. The first step to successful CPC separation is the selection of the appropriate biphasic solvent system. Based on previous work concerning the CPC fractionation of non-polar plant extract [16,17,18], several non-aqueous biphasic systems consisting of the solvents n-heptane, acetonitrile, butanol, n-hexane, isopropanol, ethyl acetate, in various combinations and proportions were tested. The initial TLC tests of these systems revealed that the biphasic system n-hexane/acetonitrile/isopropanol in the solvent ratio of 1.6/1.6/0.2 *v/v/v* showed the best distribution of the main components of the non-polar extract of *P. veris* subsp. *veris*. The initial results were verified by HPLC-UV-DAD analysis where the distribution coefficients (Kd) of the main compounds were calculated. Three hundred milligrams (300 mg) of the crude extract were fractionated in a 200 mL semi-preparative CPC column using the selected biphasic system in elution extrusion mode and reverse phase separation. The collected fractions were initially checked by TLC analysis and those of similar composition were pooled, finally resulting in 19 combined fractions (Appendix A). The TLC analysis of the combined fractions revealed a successful fractionation of the analyzed extract as well the emergence of other secondary metabolites in addition to fatty compounds.

All collected CPC fractions were submitted to UPLC-HRMS. The analysis was run in both positive and negative mode while the identification of components was guided by literature data and based on suggested elemental composition (EC), RDBeq values, and HRMS/MS spectra. The aforementioned analysis led to tentative identification of 58 metabolites, which mainly belong to five chemical categories, i.e., terpenes, flavonoids, bisbibenzyls, tocopherols/tocoenols and phenylalcanoids. Appendix A summarizes the detected compounds, their Rt, EC, the pseudo-molecular ions (*m/z*) and RDBeq value. It is important to note that flavonoid aglycons were found to be the largest categories of compounds in *P. veris* non-polar extract. Thirty lipophilic flavones with different degrees of methoxylation were identified in CPC fractions. These bioactive compounds are known components of genus Primula [19,20] and have also been identified from *P. veris* aerial parts [16].

### 2.7. In Silico Prediction of Molecular Pathways That May Intersect the Cellular Effects of Primula veris Non-Polar Extract on Influenza Virus Infection

To obtain preliminary indications of the mechanism by which non-polar extracts of Primula may impact influenza infection, we explored FLAME [21], a web tool for functional and literature enrichment analysis of multiple data sets, and DARLING, which identifies disease–bio-entity associations by performing literature mining [22].

DARLING was used to query PubMed for genes/proteins associated with the chemical terms (metabolite families) “flavone(s)”, “terpene(s)”, “bibenzyl(s)”, and “tocopherol(s)”, which were identified in the non-polar extracts of *P. veris* subsp. *veris* by UPLC-HRMS (Appendix A). This query retrieved 2133 gene/protein and miRNA terms (Appendix A). Using the same platform, we queried “Influenza” in DisGeNET [23] and retrieved 1800 PubMed ID entities containing 1251 gene/protein terms (Figure 7A and Appendix A). These genes have been positively or negatively associated with several aspects of cellular or organismal response to influenza virus infection. The intersection of “influenza” and “metabolite families” datasets was extracted in FLAME and was found to entail 571 gene/protein terms (Figure 7A and Appendix A), which were further analyzed for KEGG and Reactome [24] pathway enrichment on the FLAME platform.

KEGG analysis suggested enrichment for several viral diseases (influenza, hepatitis B and C, Kaposi sarcoma-associated herpesvirus, human cytomegalovirus infection, and EBV infection), inflammatory pathways, apoptosis, and pathways in cancer (Appendix A). Clustering of the ten most-enriched Reactome molecular pathways (Appendix A) identified three major pathway clusters that may be affected by both influenza and Primula metabolite families: a cell death/cytolysis-related cluster, the inflammasome pathway and a signaling pathway that involves p90 ribosomal S6 kinase (RSK) and its downstream target CREB (Figure 7B). Transcription factors predicted to function upstream of the intersection of Primula and influenza-associated gene terms indicated the influence of RelA/NF-κB (Appendix A).

## 3. Discussion

Data presented herein demonstrate anti-influenza virus activities in non-polar *P. veris* subsp. *veris* extracts that entail both direct virucidal effects and protective properties in the infected cells. Non-polar, DCM-derived Primula extracts have recently been shown to possess anti-fungal properties [10], but anti-influenza virus effects have not been ascribed to Primula before. Additionally, we have shown that Primula, Cistus, and Echinacea combination treatment results in amplified protection against virus-induced cytolysis and reduced progeny virus production.

Our analysis of the chemical composition of non-polar *P. veris* extracts has identified several phytochemicals with potential or proven anti-influenza properties. Among them, flavones are of particular interest given their documented capacity to treat respiratory tract infections in humans [25]. Flavone derivatives attenuate inflammatory responses to influenza virus infection in vitro [26] and inhibit H1N1 neuraminidase activity [27]. We also note that specific flavones identified in the DCM extract of Primulaceae have been reported to confer potent antifungal activity [10], along the lines of the anti-viral effects of Primula DCM extract described herein (Figure 3). Bisbibenzyl derivatives, including riccardin C, have also demonstrated antifungal properties in vitro [28] but anti-viral effects have not been ascribed to them yet. Stigmasterol, a phytosterol identified in the non-polar extracts of Primula (Appendix A) has been reported to possess anti-HA binding activity in a phytosterol extract isolated from *Erythrostemon yucatanensis,* having a direct effect on influenza virus particle infectivity [29]. The anti-influenza virus effects of loliolide have not been addressed, but this monoterpenoid lactone has been reported to inactivate free hepatitis C virus particles, to abrogate viral attachment, and to impede viral entry/fusion [30]. Alpha-tocopherol augments the therapeutic effects of oseltamivir in virus-infected mice [31] through a mechanism that may partly involve inhibition of influenza HA activity [32]. Methyl linoleate which represents a considerable percentage of fatty compounds in the esterified non-polar extract of Primula (Appendix A), has been reported to confer neuraminidase-inhibiting activity [33]. Overall, the aforementioned phytochemicals provide putative explanations for both the virucidal effects of non-polar Primula extracts and their protective properties in infected cells reported herein. Additional studies to identify specific non-polar components of Primula affecting the various stages of virus infection cycle are thus warranted.

We explored bioinformatics-based data mining to uncover putative molecular targets of Primula that may explain its protective effects on virus-infected cells. Computational strategies are promising alternatives to experimental research that have facilitated the identification of both cellular targets of influenza virus and potential therapies in the framework of “network pharmacology” [34]. By retrieving gene/protein terms associated with both influenza and key phytochemicals found in non-polar Primula extract, we identified cell death-related pathways, the NLRP3 inflammasome, and RSK/CREB signaling as potential cellular targets of the medicinal plant extracts pertinent to virus infection (Figure 7). Indeed, influenza virus proteins interfere with components of both the intrinsic and extrinsic cell death pathways that include TRAIL, TNF, and Fas ligands, triggering activation of caspase-8 and/or caspase-3 [35,36]. Primula extracts are predicted to inhibit cytolysis by interfering with these death-inducing pathways. Among its components, flavonoids are putative effectors for survival given their reported capacity to ameliorate the cytopathic effects of enterovirus 71 [37].

The NLRP3 inflammasome is a multiprotein intracellular complex comprising NLRP3, ASC, and pro-caspase-1 that is activated upon infection with influenza [38], as well as other RNA viruses [39]. Inflammasome activation results in RelA/NF-κΒ signaling and the production of interleukin 1β (IL-1β), IL-18, and several other inflammatory cytokines. 5-Methoxyflavone, which was identified in non-polar Primula extracts, inhibits NF-κB and NF-κB-dependent inflammatory molecule production in response to influenza A virus infection [26]. Moreover, suppression of NF-κB by either small molecule inhibitors [40] or polyphenol/flavone-rich natural products [41,42] attenuates influenza virus replication. Likewise, inhibition of virus-induced activation of CREB by geniposide, an iridoid glycoside purified from the fruit of *Gardenia jasminoides J.Ellis*, has been associated with reduced neuraminidase activity [43]. Thus, we envisage that modulation of the NF-κB and CREB transcription factors by Primula phytochemicals may contribute to protection from the cytolytic effects of the virus, possibly through various routes. Further studies are warranted to provide functional evidence of these in silico-based hypotheses.

Overall, we have herein uncovered novel anti-influenza virus properties of dichloromethane and supercritical CO_2_/ethanolic extracts of *Primula veris* subsp. *veris* in vitro and an amplified prophylactic and therapeutic effect when combined with hydroethanolic extracts from *C. creticus* subsp. *creticus*. These plants, together with *E. purpurea,* have been used in traditional medicine in southern Europe for centuries without any reported complications. In vivo, this triple combination may confer systemic anti-inflammatory and immune-boosting effects, in addition to direct anti-viral properties [44,45,46], warranting further studies to assess clinical efficacy.

## 4. Materials and Methods

### 4.1. Chemicals and Reagents

The methanol, ethanol, isopropanol, dicloromethane, n-hexane, ethyl acetate, acetonitrile, water, and acetic acid used for the extractions, as well as the CPC chromatography and HPLC analyses were of analytical grade and purchased from Fisher Scientific (Pittsburgh, PA, USA). LC-MS grade acetonitrile, water, and formic acid (Fisher Scientific, Pittsburgh, PA, USA) were used for UPLC-HRMS/MS analyses. Deuterated methanol and deuterated chloroform were used for 1H NMR experiments.

### 4.2. Plant Material

The flowers of *P. veris* subsp. *veris* (Primulaceaea) were collected from Fourka region of Ioannina on 16 May in 2019. The aerial parts of *C. creticus* subsp. *creticus* (Cistaceae) were collected near the village of Sises, Rethymno, Crete on 3 May 2021. The plant materials were authenticated by the botanist Dr. Eleftherios Kalpoutzakis and a voucher specimen (A-LS 020 for the sample of *P. veris* subsp. *veris* and KL587 for the sample of *C. creticus* subsp. *creticus*) was deposited in the herbarium of the Division of Pharmacognosy and Natural Products Chemistry, Department of Pharmacy, National Kapodistrian University Athens, Greece. The plant material of *E. purpurea* (Asteraceae) was purchased from local markets in Athens, Greece. All plant materials were dried under shadow and pulverized by a ground mill.

### 4.3. Extraction

The extraction of *P. veris* subsp. *veris* took place using two different techniques, the ultrasound extraction (UE) and supercritical fluid extraction (SFE). For UE experiments 200 g of the dry material were added into a laboratory scale device of 3 L volume (R.E.U.S^®^, Contes, France) and extracted initially with 1.5 L dichloromethane (DCM extr.), then with 1.5 L methanol (MeOH extr.) and finally with 1.5 L of water (H_2_O extr.). Each extraction lasted 30 min, the supernatants were filtrated and evaporated under vacuum, finally yielding 3.64 g of DCM extract, 22.44 g of MeOH extract, and 32.33 g of H_2_O extract. The SFE experiments were run in a Separex^®^ SFE100 device (Champigneulles, France). Twenty grams of dry powdered material were initially extracted with supercritical CO_2_ (CO_2_ extract) and then by adding ethanol as a co-solvent in ratios of 5% (CO_2_ + 5% EtOH extract) and 13% (CO_2_ + 13% EtOH extract). Each step of the extraction procedure lasted 90 min, resulting in the recovery of 110 mg of CO_2_ extract, 306.9 mg of CO_2_ + 5% EtOH extract, and 149.1 mg of CO_2_ + 13% EtOH extract. The other experimental parameters were as follows: flow rate of CO_2_ at 25 g/min, pressure level at 250 bar, extraction temperature at 40 °C, and separator temperature at 35 °C.

The UE technique was applied for the extraction of *C. creticus* subsp. *creticus* and *E. purpurea* plant materials in laboratory scale. Two batches of 20 g of the powdered row material of *C creticus* subsp. *creticus* were extracted with 250 mL of ethanol/water 1/1 and 250 mL of water, respectively. Each extraction procedure lasted 30 min and the collected eluents were evaporated to dryness, recovering 5.16 g of hydroalcoholic extract (25.8% *w/w*) and 4.54 g of aqueous extract (22.7% *w/w*). Regarding the *E. purpurea* aerial part, 20 g of powdered material was initially extracted successively with dichloromethane, methanol and methanol/water 1/1. The collected eluents were evaporated to dryness and weighed, yielding 0.197 g of DCM extract (0.98% *w/w*), 0.667 g of MeOH extract (3.34% *w/w*), and 1.72 g of MeOH/H_2_O extract (8.6% *w/w*). Then, another batch of 20 g of powdered material was directly extracted with EtOH/H_2_O (1/1 *v/v*), the eluent was evaporated to dryness and weighed, yielding 2.24 g of dry extract (11.2% *w/w*). Moreover, a part of the EtOH/H_2_O extract was treated with absorption resin Amberlite XAD-7 and the phenolic fraction was recovered.

### 4.4. Fractionation of P. veris subsp. veris Non-Polar Extract

Countercurrent partition chromatography (CPC) was applied to fractionate the DCM extract of *P. veris* subsp. *veris*. The experiment was carried out using an FCPC^®^ Kromaton device (France) equipped with a 200 mL semi-preparative CPC column. The solvent flow was carried out by means of a LabAlliance pump and the fractions were collected using a Buchi B684 automatic collector. The separation was achieved using the biphasic system n-hexane/acetonitrile/isopropanol 1.6/1.6/0.2 *v/v/v* in elution extrusion mode. The analysis started by filling the column with the upper phase of the biphasic system (stationary phase) at a flow rate of 20 mL/min and 200 rpm. Then, the lower phase (mobile phase) was passed through the column at a flow rate of 10 mL/min and column rotation speed of 800 rpm. After the hydrodynamic equilibrium of the two phases into the column (Sf = 65%) the sample was injected (300 mg of the extract dissolved in 10 mL of a mixture of the two phases). The flow rate and rotation speed were set at 10 mL/min and 800 rpm for the entire procedure while the fraction collector was set to collect fractions every 1 min. Using the lower phase as a mobile phase (elution step), 60 fractions of 10 mL were collected. Then the upper phase was set as mobile phase (extrusion step) and 35 more fractions of 10 mL were collected. All obtained fractions were subjected to TLC analysis and pulled based on their chemical similarity, thus resulting in 19 combined fractions.

### 4.5. TLC and HPLC Analysis

TLC analysis was performed on silica gel 60 F254 and silica gel RP-18 F254S TLC aluminum plates (Merck, Darmstadt, Germany). For the normal phase, the chromatographs were developed in a solvent systems consisting of dichloromethane and methanol in ratio 100/0, 98/2, and 90/10 (*v/v*) and for the reverse phase the chromatographs were developed in a solvent system consisting of H_2_O-AcN 70:30 (*v/v*). Spots were visualized at 254 and 366 nm and at visible after treatment with a sulfuric vanillin solution (5% *w/v* in methanol)—H_2_SO_4_ (5% *v/v* in methanol) and followed by heating at 120 °C for 1 min.

The qualitative HPLC analysis was performed in a Termo Finnigan device consisted of a SpectraSystem P4000 pump, a SpectraSystem 1000 degasser, a SpectraSystem AS3000 auto sampler and SpectraSystem UV6000LP detector. The chromatographic separation was performed on a Supelcosil LC-18 (25 cm × 4.6 mm i.d., 5.0 μm) column (Merck) using as mobile phase mixtures of acidified water (1% acetic acid) (solvent A) and acetonitrile (with 2% methanol and 1% acetic acid) (solvent B) in the following gradients for the aqueous extract: from 0 to 10 min, 95% (A) to 85% (A); from 10 to 45 min, 85% (A) to 75% (A); from 45 to 52 min, 75% (A) to 5% (A); from 52 to 53 min, 5% (A) to 95% (A); from 53 to 55 min, 95% (A) to 95% (A). The flow rate was set at 1 mL/min, and the injection volume at 10 μL. The detection of the eluted compounds was performed using a PDA detector. For the monitoring of the runs 254, 280, and 355 nm were used simultaneously.

### 4.6. GC-MS and UPLC-HRMS Analysis

The GC-MS analyses of crude and esterified DCM extracts of *P. veris* were performed on an Agilent Technologies 7820A apparatus coupled to an Agilent Technologies 5977B MSD mass spectrometer. The nonpolar compounds were separated in an HP-5MS capillary column (30 m 0.25 mm; film thickness of 0.25 m Agilent Palo Alto, CA, USA.) using Helium as carrier gas at a flow rate of 1 mL/min. The gas chromatograph oven temperature started at 60 °C and increased by 3 °C per minute until it reached 280 °C where it remained for 10 min. The injected volume was 1 μL and the total analysis for each sample lasted 83.3 min. The ion production method was an electron bombardment (EI-70 eV). The identification of the compounds was based on a comparison of the obtained MS spectra with the mass spectra of the Adams, Nist, and Wiley libraries

An H-Class Acquity UPLC system (Waters, Milford, MA, USA) coupled to an LTQ-Orbitrap XL hybrid mass spectrometer (Thermo Scientific, Waltham, MA, USA) was used to perform UPLC-HRMS/MS analyses of the CPC fractions. The chromatographic separation was carried out on a Supelco C18 (15 × 2.1 mm, 3 μm) column at a temperature of 40 °C, using as mobile phase mixtures of acidified water (0.1%formic acid) (solvent A) and acetonitrile (solvent B) in the following gradient: from 0 to 2 min, 98% (A) and 2% (B), from 2 to 18 min, 98% (A) and 2% (B), from 18 to 21 min 0% (A) and 100% (B), from 21 to 22 min 0% (A) and 100% (B), from 22 to 25 min 98% (A) and 2% (B). The flow rate was set at 0.4 mL/min and the injection volume at 10 μL. The HRMS data were acquired in positive and negative mode with a mass range of 115–1000 *m/z*. The ESI conditions were as follows: capillary temperature 350 °C; capillary voltage −30 V for negative and 40 for positive mode; tube lens −100 V for negative and 120 for positive mode; sheath gas 40 Au; auxiliary gas 10 Au; sweep gas 0. The raw data were acquired and processed with Xcalibur 2.0.7 software (Thermo Scientific, Waltham, MA, USA).

### 4.7. Cell Culture, Treatments and Cytotoxicity Assays

Madin–Darby canine kidney (MDCK) cells are frequently used for the primary isolation of influenza viruses and assessment of virus-induced cytopathic effects because of their high susceptibility to infection with various influenza strains [47,48]. The dried extracts were dissolved in DMSO and further diluted in medium prior to cell treatment. For the assessment of survival following treatment with plant extracts, 5000 MDCK cells were plated per well in a 96-well plate and the next day increasing amounts of extracts were applied over a culture period of 72 h. Viability was assessed by MTT conversion assays according to a standard methodology [49]. Alternatively, we assessed cell lysis was assessed by quantifying the release of lactate dehydrogenase (LDH) using the CytoTox 96^®^ assay according to the manufacturer’s instructions (Promega), using the respective amounts of DMSO as negative control. MDCK cells were provided by Dr V. Andreakos (Biomedical Research Foundation of the Academy of Athens, Greece) [50].

### 4.8. Virus and Infections

Human influenza virus A/Puerto Rico/8/34 (H1N1/PR8) was kindly provided by Dr V. Andreakos (Biomedical Research Foundation of the Academy of Athens, Greece) [50]. For infection, MDCK cells were grown in 96-well plates for 24 h. The cells were inoculated with H1N1 PR8 virus at 0.1 or 0.2 MOI using DMEM containing 0.15 g/mL trypsin and 5 μg/mL bovine serum albumin (BSA) (Sigma, Burlington, MA, USA). After 1 h of incubation, the cells were washed with phosphate-buffered saline (PBS) (Gibco) and the medium was replaced with new medium. The cells were continually incubated for 24–30 h at 37 °C under a 5% CO_2_ atmosphere.

### 4.9. Time-of-Addition Assays

For the quantification of virus-induced cytolytic effects and assessment of the inhibitory effects of plant extracts on influenza virus, 3 modes of treatment were explored (Figure 1): (a) H1N1 virus was incubated with 50 μg/mL of each extract for 1 h prior to infection of MDCK cells at MOI = 0.1 (virucidal protocol); (b) MDCK cells were treated with 50 μg/mL of each plant extract for 4 h, washed, infected with H1N1 and then cultured in standard growth media (prophylactic protocol); (c) MDCK cells were infected with H1N1 and then cultured in the presence or absence of 25 or 50 μg/mL of each plant extract (“therapeutic” protocol).

### 4.10. Plaque Titrations

Supernatants, collected at 24 h post-infection, were used to assess the number of infectious particles (plaque titers) in the sample. Briefly, MDCK cells grown to 90% confluency in 6-well dishes were washed with PBS and infected with serial dilutions of the supernatants in PBS supplemented with 0.2% BSA for 30min at 37 °C. The inoculum was aspirated, and cells were incubated with 2 mL DMEM supplemented with 0.2% BSA, 0.6% Agar (Oxoid), 0.3% DEAE-Dextran (Pharmacia Biotech), and 1.5% NaHCO_3_ at 37 °C, 5% CO_2_ for 2–3 days. Virus plaques were visualized by staining with neutral red.

### 4.11. In-Silico Predictions of Pathways That May Mediate the Anti-Viral Effects of Primula

In silico predictions were performed by combining FLAME [21] and DARLING [22] platforms. DARLING enables disease–bio-entity associations with literature mining and FLAME is a web tool for functional and literature enrichment analysis of multiple data sets. The former tool was used to query PubMed for chemical terms (bio-entities) found in Primula and to query DisGeNET for terms (genes/proteins) associated with “Influenza”. FLAME was next utilized to identify their intersection and perform pathway enrichment analyses as described in the Results section.

## 5. Conclusions

In summary, we have uncovered anti-influenza virus activity in non-polar *P. veris* subsp. *veris* extracts that entails both direct virucidal effects and protective properties in the infected cells. We propose that non-polar *P. veris* alone or in combination with extracts of medicinal plants with proven anti-influenza activity, such as *Cistus creticus* and *Echinacea purpurea* may be particularly useful in protecting against the clinical manifestations of influenza in humans. Future studies will address this hypothesis and identify the *P. veris* phytochemicals responsible for inhibition of the various stages of virus cycle.

## 6. Patents

A patent application resulting from the work reported in this manuscript has been submitted to the Hellenic Industrial Property Organisation with registration number 20220100185.

## Figures and Tables

**Figure 1 pharmaceuticals-15-01513-f001:**
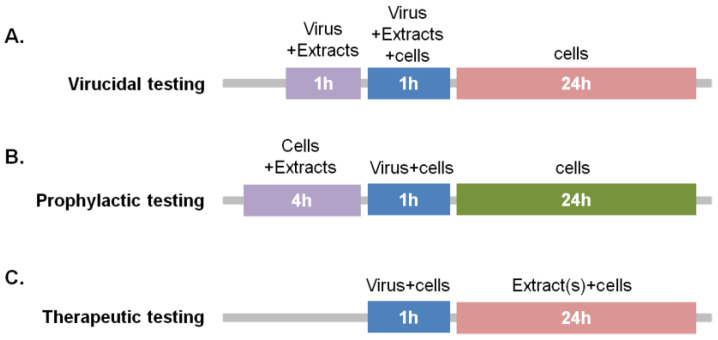
Schematic representation of the administration protocols used in this study. (**A**) Virucidal activity was tested by incubating the virus with plant extract for 1 h, prior to MDCK cell infection, as indicated. (**B**) Prophylactic testing was performed by pre-incubating MDCK cells with plant extract, infecting with H1N1 virus and measuring cell lysis 24 h post-infection. (**C**) Therapeutic activity was tested by infecting cells with the virus for 1 h and then applying the plant extract for 24 h prior to assessment of cell lysis by LDH release.

**Figure 2 pharmaceuticals-15-01513-f002:**
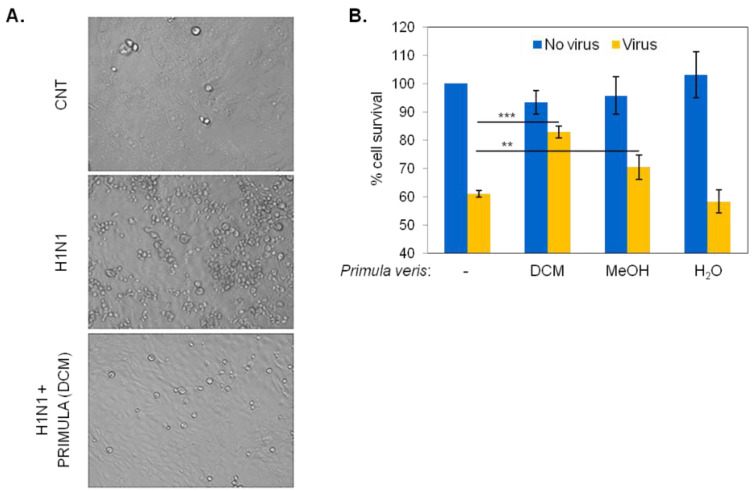
Virucidal effects of *P. veris* non-polar (DCM), medium-polarity (MeOH) and polar (H_2_O) extracts. (**A**) Microphotographs of MDCK cells infected with H1N1 (MOI = 0.1) or with virus that had been pre-incubated for 1 h with 50 μg/mL of *P. veris* non-polar (DCM) extract. Rounded cells are indicative of virus-induced cytotoxicity. CNT: uninfected control cultures. (**B**) Collective data of MDCK cell survival, assessed by MTT conversion assays, following infection with either H1N1 alone or *P. veris* extract-treated H1N1, as described in (**A**). Data are the average of duplicate determinations from 3 independent experiments. (*** *p* < 0.001; ** *p* < 0.01 Student’s *t*-test).

**Figure 3 pharmaceuticals-15-01513-f003:**
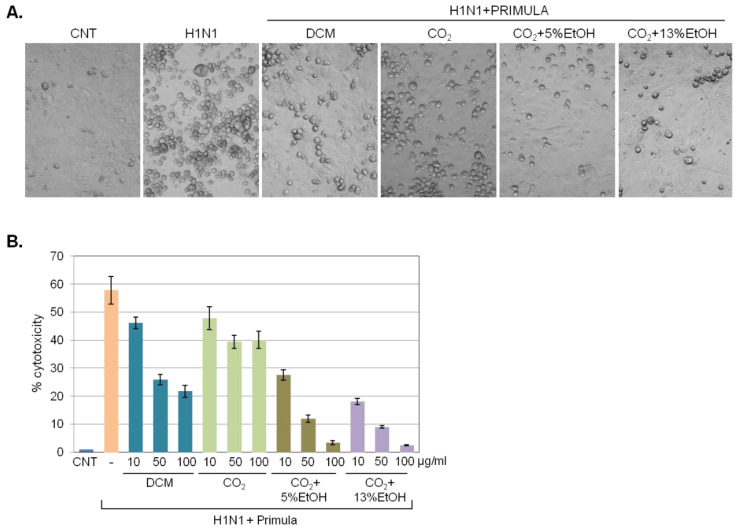
Virucidal effects of *P. veris* supercritical extracts. (**A**) Microphotographs of MDCK cells infected with H1N1 (MOI = 0.2) or with virus that had been pre-incubated for 1 h with 50 μg/mL of *P. veris* extract isolated by using supercritical CO_2_ in the presence or absence of 5% or 13% EtOH. As control, DCM extracts of *P. veris* were used. Rounded cells are indicative of virus-induced cytopathic effects. (**B**) MDCK cells were infected with H1N1 (MOI = 0.2) or with virus that had been pre-incubated for 1 h with 10, 50 or 100 μg/mL of *P. veris* extracts, as indicated. Virus-induced cytolytic effects were assessed by LDH release assays. CNT: uninfected control cultures.

**Figure 4 pharmaceuticals-15-01513-f004:**
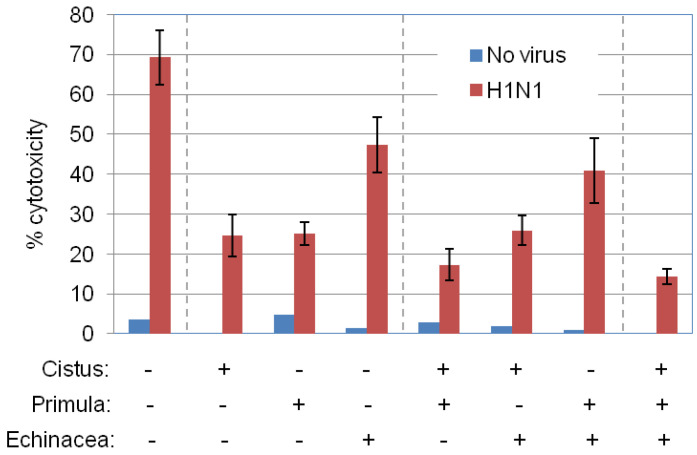
Prophylactic effects of *P. veris* on influenza virus cytolysis. MDCK cells were incubated for 4 h with 50 μg/mL Primula extracts isolated by supercritical CO_2_ + 13% EtOH, and then infected with H1N1 at MOI = 0.2 according to the protocol of Figure 1B. Echinacea (EtOH/H_2_O-Xad7-derived) and Cistus (hydroethanolic) extracts were used in parallel for comparison, in the presence or absence of Primula extract. The graph depicts virus-induced MDCK cell lysis, assessed by LDH release assays. Data are the average of duplicate determinations from 3 independent experiments.

**Figure 5 pharmaceuticals-15-01513-f005:**
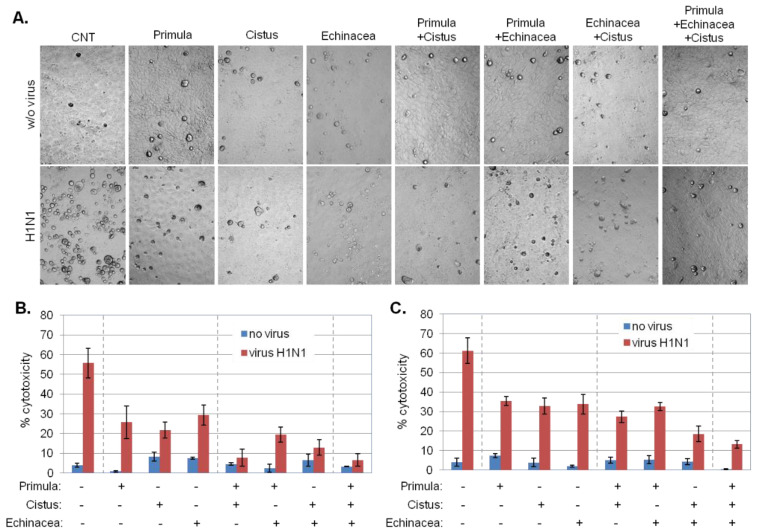
Therapeutic effects of *P. veris* extracts on influenza virus-induced cytolysis. (**A**) Microphotographs of MDCK cells infected with H1N1 (MOI = 0.2) followed by 24 h treatment with 50 μg/mL Primula (supercritical CO_2_ + 13% EtOH), Echinacea (EtOH/H_2_O-Xad7-derived), Cistus (hydroethanolic) extracts and their combinations. Rounded cells are indicative of virus-induced cytotoxicity. CNT: control cultures which were either remained uninfected (w/o virus) or infected with H1N1 in the absence of further treatment. (**B**,**C**) Collective data of virus-induced MDCK cell lysis, assessed by LDH release assays, following infection with either H1N1 in the presence or absence of 50 μg/mL (**B**) or 25 μg/mL (**C**) Primula, Echinacea, Cistus extracts and their combinations, as indicated. Data are the average of duplicate determinations from 3 independent experiments.

**Figure 6 pharmaceuticals-15-01513-f006:**
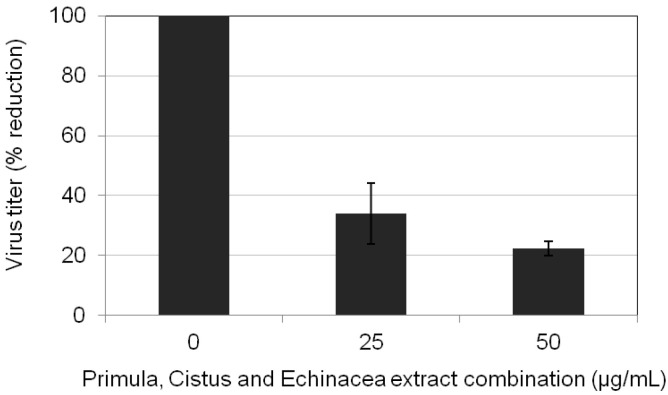
Triple combination of Primula (supercritical CO_2_ + 13% EtOH), Cistus (hydroethanolic), and Echinacea (EtOH/H_2_O-Xad7-derived) extracts reduce influenza virus propagation. MDCK cell cultures were infected with H1N1 at MOI = 0.01 in the absence or presence of 25 μg/mL or 50 μg/mL each plant extracts before, during, and after infection. Supernatants from each infected culture were assayed for progeny virus yields at 24 h post-infection by measuring plaque formation. Virus titers (plaque forming units/mL) of untreated cells were 1.3 (±0.2) × 10^3^ pfu/mL and were arbitrarily set as 100%.

**Figure 7 pharmaceuticals-15-01513-f007:**
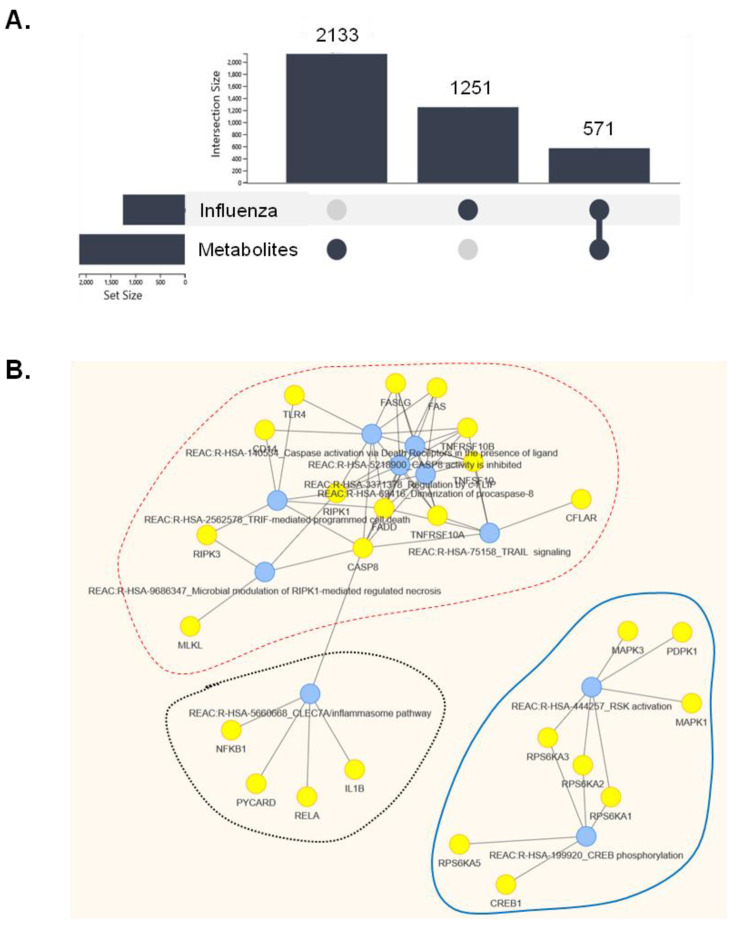
Bioinformatics-based prediction of pathways that intersect Primula metabolites and influenza-associated gene/protein terms. (**A**) Graph showing the number of genes/proteins associated with influenza (n = 1251), with phytochemicals that have been identified in Primula (metabolites, n = 2133), and their intersection (n = 571). (**B**) Enrichment of the intersection gene terms identified in (A) for Reactome molecular pathways using data p value correction cut-off 0.01. The ten most-enriched pathways and their interactions are shown. Three main clusters were identified: a cell death/cytolysis-related cluster (framed with dotted red line), inflammasome pathway (circled by a black dotted line) and an RSK/CREB-related signaling pathway (circled by blue line).

## Data Availability

Data is contained within the article and Appendix A.

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
