# Peer review of "In Vitro Anti-Influenza Virus Activity of Non-Polar Primula veris subsp. veris Extract"

_pharmaceuticals, 2022, doi:10.3390/ph15121513_

Round 1

Reviewer 1 Report

The study by Eliopoulos et al. has tried to address an important issue by providing a green alternate to currently prevailing antivirals against influenza virus. The study is indeed important as far as the translational aspect is considered, however, has numerous flaws as several controls are missing. 

Some of the issues are:

1. Please check the typological errors as in

Line 81: Check the spelling of "raw material'.

Line No, 83: Put commas before and after "respectively'.

2. Throughout the study, one major drawback is the absence of negative control. Although authors have used an antiviral as the positive control, but havent used any negative control. The solvents used to prepare the plant extract are also known to possess antiviral activity and have the potential of neutralizing the virus. So the antiviral activity of all the solvents included in study should be assessed, along with their MTT assays.

3. The authors have tried to claim that the plant extract is selectively killing the virus infected cells but havenot provided any mechanism for the same. How the plant extract is selectively able to distinguish between infected and non-infected cells? Please explain? Which pathway is it targeting in the influenza virus life cycle?

4. In line 161, the authors have mentioned that they have used 13% ethanol for the preparation of plant extract and in line no 169, authors do mention the treatment for upto four hours with the plant extract. Thirteen percent concentration of ethanol is quite toxic for any cell line and can cause enough dehydration that can lead to cell death. Here also, the negative control is missing. Authors should include a control here to support their statement. Pretreatment of the cells for 4 hours followed by the assessment of percent viability after 24 hours post treatment can give a better idea.

5. Please improve the quality of Figure 6.

6. With the discovery of any new drug, economy of the whole process has always been a question of interest that further governs its practical admissibility? What was economy of whole process?

Reviewer 2 Report

The article is original, well structured; easy to read with main emphasis on the effects of non-polar Primula veris extract against anti-influenza virus. Why the author has mentioned extracts in title. As per the manuscript, it is clearly evident that the author has only used a single extract. If not, then please reframe the title. The abstract is well presented with all the significant research findings. The statistical significance of data is also very well explained in the figure captions. The authors have explained the methodology section in an elaborated manner which is appreciable. Also, the results are very well explained and correlated. These results are strongly supporting the theme of the research. In my opinion, the manuscript can only be reconsidered after the incorporation of these suggestions:

1.      Line 41: illness spelling is wrong.

2.      Line 66-67: Sentence is not clear

3.      Please rewrite section 2.1 and don’t write steps 1 and step 2 in brackets.

4.      Try to add a few lines in the conclusion section mentioning the futuristic scope of the research.

5.      Also, there are several places in the introduction and discussion section, where the author must reframe the sentences.

6.      I would suggest the potential author carefully examine the introduction section.

Round 2

Reviewer 1 Report

Dear Editor and Authors

The changes are seeming fine to me. It can be accepted for publication.